# AIR: Zero-shot Generative Model Adaptation with Iterative Refinement

## Abstract

Zero-shot generative model adaptation (ZSGM) aims to adapt a pre-trained generator to a target domain using only text guidance and without any samples from the target domain. Central to recent ZSGM approaches are *directional loss* which use the text guidance in the form of aligning the image offset with text offset in the embedding space of a vision-language model like CLIP. This is similar to the analogical reasoning in NLP where the offset between one pair of words is used to identify a missing element in another pair by aligning the offset between these two pairs. However, a major limitation of existing ZSGM methods is that the learning objective assumes the complete alignment between image offset and text offset in the CLIP embedding space. **Our work** makes two main contribution. Inspired by the offset misalignment studies in NLP, as our first contribution, we perform an empirical study to analyze the misalignment between text offset and image offset in CLIP embedding space for various large publicly available datasets. Our important finding is that offset misalignment in CLIP embedding space is correlated with concept distance, *i.e.*, close concepts have a less offset misalignment. To address the limitations of the current approaches, as our second contribution, we propose Adaptation with Iterative Refinement (AIR) which mitigates the offset misalignment issue in directional loss by iteratively selecting anchor points closer to the target domain. Extensive experimental results show that the proposed AIR approach achieves SOTA performance across various adaptation setups. **Code and additional experiments in Supp**.

## 1 Introduction

Generative models like Generative Adversarial Networks (GANs) Goodfellow et al. (2014); Karras et al. (2019; 2020b); Brock et al. (2019) and Diffusion Models Rombach et al. (2022); Nichol & Dhariwal (2021); Dhariwal & Nichol (2021) have recently shown promising results in image generation with significant advancements in the fidelity and diversity of the generated images. Training these generative models typically requires large amounts of data (*e.g.*, 70K images required for training StyleGAN2 Karras et al. (2020c) or 400M images used for training Latent Diffusion Model Rombach et al. (2022)). However, in many real-world scenarios, a limited amount of data is available from the target domain (*e.g.*, medical domains, rare animal species, and artistic domains). Training a generative model under this limited data regime is extremely challenging, resulting in issues like mode collapse or quality degradation Abdollahzadeh et al. (2023). To address this, *generative model adaptation* uses transfer learning to adapt a generator pre-trained on a similar, well-represented domain to a new target domain with limited data Li et al. (2020); Zhao et al. (2022a;b; 2023). Leveraging the recent advances in vision language models like CLIP Radford et al. (2021), takes this one step further and enables the *zero-shot generative model adaptation* (ZSGM) which shifts a pre-trained model to a target domain using only text guidance (no images from target domain).

**ZSGM with Offset Alignment.** NADA Gal et al. (2022) is the pioneering work that uses text offset between source and target domains in CLIP embedding space as guidance to shift a pre-trained generative model to the target domain. Specifically, the learning objective is to align the image offset between the pre-trained generator and the adapted generator with the text offset. IPL Guo et al. (2023) improves on this by using prompt learning to enhance diversity, addressing NADA's

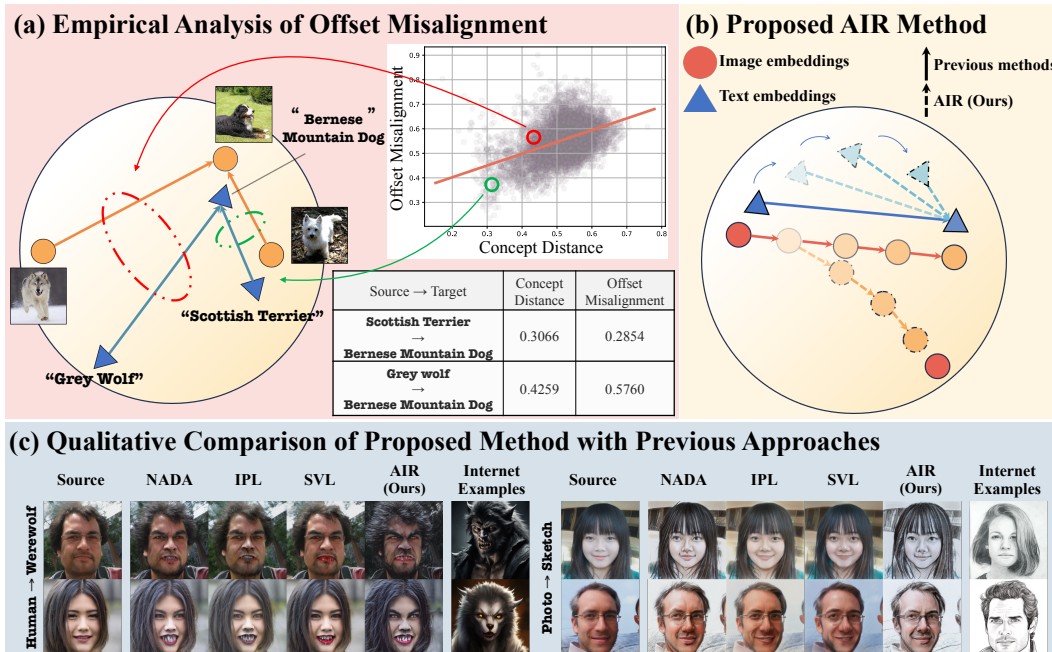

Figure 1: **Overview and our contributions: (a)** We perform an empirical analysis of the offset misalignment in CLIP embedding space. Our analysis reveals that there is a misalignment between image offset (orange arrow) and text offset (blue arrow) in CLIP space, and this misalignment is correlated with concept distance. For example, in the ImageNet-1K dataset, the "Gray Wolf" is a more distant concept to the "Bernese Mountain Dog" (concept distance=0.4259) than the "Scottish Terrier" (concept distance=0.3066). Therefore, considering "Bernese Mountain Dog" as the target, the "Gray Wolf" has more concept misalignment (0.5760 vs 0.2854). This misalignment is ignored in the existing approaches which results in sub-optimal learning with directional loss (See Sec. 3). **(b)** We propose Adaptation with Iterative Refinement (AIR) to mitigate the issue with offset misalignment. We iteratively sample anchor points closer to the target domain and use these anchors to refine the adaptation direction (Sec. 4). **(c)** Our proposed AIR consistently archives SOTA performance across different adaptation setups (Detailed quantitative and qualitative results in Sec. 5).

limitations in image-specific feature representation. SVL Jeon et al. (2023) advances further by modeling semantic variations to tackle mode collapse. The idea of offset alignment has some similarities to analogical reasoning in NLP literature Mikolov et al. (2013c;a;b); Levy & Goldberg (2014) where the offset between one pair of word vectors is used to identify the unknown member of a different pair of words, commonly via alignment of offsets. For example the offset $E_v$("Man") - $E_v$("Woman") and $E_v$("King") being used to identify $E_v$("Queen"), with $E_v$ denoting the vector representation in some embedding space. See Sec. G for detailed discussion on related work.

**Research Gap.** All current ZSGM approaches Gal et al. (2022); Guo et al. (2023); Jeon et al. (2023) assume that *the image and text offsets are completely aligned in the CLIP embedding space* and leverage this in their learning objective while adapting the pre-trained generator to the target domain. However, this assumption can have two major limitations: i) CLIP embedding space is trained to maximize the similarity between corresponding image-text pairs, and the degree of alignment between image and text offsets is not studied properly, and ii) this degree of alignment could also vary based on the distance between source and target domains. Recalling the similarity between ZSGM with offset alignment and analogical reasoning, previous studies in NLP have shown that the accuracy of analogical reasoning increases if the concepts are nearby and similar (*e.g.,*, $E_v$("King") $E_v$("Queen") Levy et al. (2015); Köper et al. (2015); Vylomova et al. (2015); Rogers et al. (2017); Fournier et al. (2020)), and decreases when the concepts are distant.

**Contributions.** This paper takes an important step toward addressing the research gaps in ZSGM. *First*, we take a closer look to analyze the offset misalignment in CLIP embedding space. Specif-

ically, we perform an empirical study on large public datasets to analyze the degree of the offset alignment between image and text offsets in CLIP embedding space vs concept distance. Our results suggest that offset misalignment exists in the CLIP embedding space and it increases as the concepts are more distant (Fig. 1(a)). Additionally, we perform a set of experiments to show that for closer source and target domains, the offset misalignment is less problematic during ZSGM. ***Second***, informed by our analysis, we propose Adaptatoin with Iterative Refinement (AIR) to mitigate the offset misalignment issue. Our intuition is that after limited iterations of the adaptation, the adapted generator is already closer to the target domain than the pre-trained generator, and therefore it suffers less from offset misalignment. Then, we iteratively sample anchor points during adaptation and use these anchor points to calculate the offsets (Fig. 1(b)). Since the textual description of these anchor points is unknown, we propose a new prompt learning strategy to learn these descriptions. Our main contributions are summarized as follows:

- We conduct an empirical analysis of the offset misalignment between image and text modalities in the CLIP embedding space. For the first time in literature, our analysis reveals that the misalignment is larger for distance concepts and less for close concepts.

- We propose the Adaptation with Iterative Refinement to address the offset misalignment in CLIP embedding space. Our approach includes an iterative sampling of anchor points during adaptation coupled with a new prompt learning approach to learn the textual description of these anchor points.

- Extensive experimental results show that our proposed AIR approach consistently outperforms existing ZSGM approaches achieving new SOTA performance. We remark that for the first time in the literature, we perform zero-shot adaptation for the diffusion models.

## 2    PRELIMINARIES: DIRECTIONAL CLIP LOSS

In zero-shot generative model adaptation setup Gal et al. (2022), given a pre-trained generator $G_\mathcal{S}$ on the source domain $\mathcal{S}$, and textual descriptions of source and target domains, denoted by $T_\mathcal{S}$ and $T_\mathcal{T}$ respectively, the goal is to shift $G_\mathcal{S}$ to target domain $\mathcal{T}$ to generate diverse and high-quality images from this domain Abdollahzadeh et al. (2023). For this adaptation, current approaches Gal et al. (2022); Guo et al. (2023); Jeon et al. (2023) use the CLIP model Radford et al. (2021) as the source of supervision, and assume that text and image offsets (between $\mathcal{S}$ and $\mathcal{T}$) are well-aligned in CLIP representation space. Therefore, the text offset is computed based on the provided textual descriptions of the source and target. Then, the trainable generator is initialized with the parameters of the $G_\mathcal{S}$, and optimized in a way to align image offset with text offset, leading to the directional CLIP loss:

$$\mathcal{L}_{direction} = 1 - \cos(\Delta I_{\mathcal{S} \to t}, \Delta T_{\mathcal{S} \to \mathcal{T}}),$$
$$\text{where } \Delta I_{\mathcal{S} \to t} = E_I(G_t(w)) - E_I(G_\mathcal{S}(w)), \qquad (1)$$
$$\text{and } \Delta T_{\mathcal{S} \to \mathcal{T}} = E_T(T_\mathcal{T}) - E_T(T_\mathcal{S})$$

where $\cos(x, y) = x \cdot y / |x||y|$ represents the cosine similarity. $E_T$ and $E_I$ denote the CLIP text and image encoders, respectively. $G_t$ denotes the trainable generator in iteration $t$ of adaptation. $\Delta I_{\mathcal{S} \to t}$ denotes the image offset computed from the source generator to the trainable generator, and $\Delta T_{\mathcal{S} \to \mathcal{T}}$ denotes the text offset from source to target.

## 3    A CLOSER LOOK AT OFFSET MISALIGNMENT IN CLIP SPACE

Previous works assume that for two different concepts $\alpha$ and $\beta$, the image offset $\Delta I_{\alpha \to \beta}$ and text offset $\Delta T_{\alpha \to \beta}$ are completely aligned in the multimodal CLIP embedding space. This assumption of perfect alignment is the foundation of the directional loss in Eq. 1. We postulate that this assumption may have two major limitations:

- CLIP Radford et al. (2021) is trained using a contrastive loss to maximize the cosine similarity between corresponding image-text pairs, *i.e.,* maximize $cos(E_I(I_\alpha), E_T(T_\alpha))$ for concept $\alpha$ (*e.g.*, cat), or maximize $cos(E_I(I_\beta), E_T(T_\beta))$ for concept $\beta$ (*e.g.*, dog). Note that the degree of alignment of image offset $\Delta I_{\alpha \to \beta}$ and text offset $\Delta T_{\alpha \to \beta}$ in CLIP space is not studied in the literature.

- In addition, this degree of alignment between $\Delta I_{\alpha\to\beta}$ and $\Delta T_{\alpha\to\beta}$ may vary based on the distance between two concepts $\alpha$ and $\beta$.

In this section, we take a closer look at this degree of offset alignment between two different modalities in CLIP space. First, inspired by offset misalignment in NLP, we conduct an empirical study on large public datasets to analyze the offset misalignment between image and text modalities in CLIP embedding space. Our analysis suggests that **there is a misalignment between $\Delta I_{\alpha\to\beta}$ and $\Delta T_{\alpha\to\beta}$ in CLIP embedding space, and this misalignment increases as concepts $\alpha$ and $\beta$ become more distant**. Second, we take a further step and design an experiment to evaluate the effect of this offset misalignment in generative model adaptation using directional loss (Eq. 1). Our experimental results suggest that **less offset misalignment in CLIP embedding space leads to a better generative model adaptation with directional loss**.

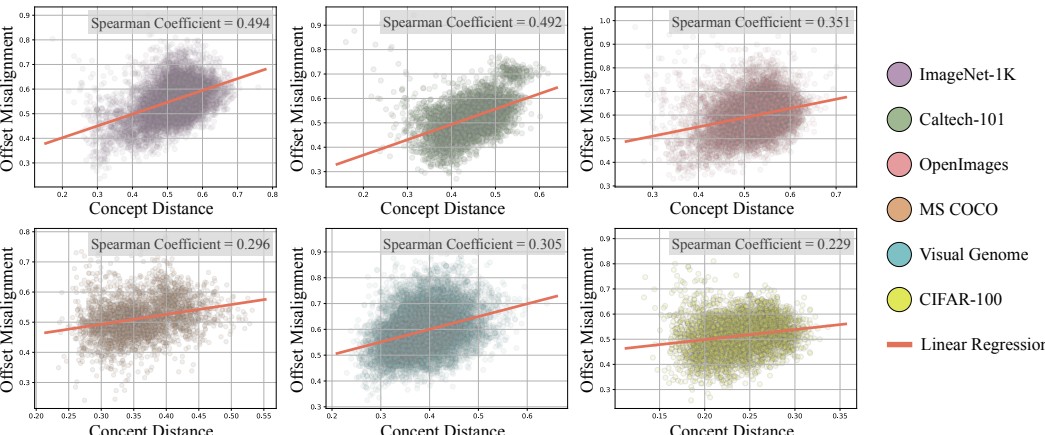

Figure 2: **Empirical analysis of offset misalignment in CLIP space:** We plot the offset misalignment (Eq. 2) vs concept distance (Sec. A.1) for $N = 5000$ of text-image pairs in CLIP space which are sampled from 6 large publicly available datasets. Our results show that there is a meaningful correlation (measured by Spearman's coefficient Zar (2005)) between offset misalignment and concept distance for datasets with different distributions, i.e., close concepts has less offset misalignment.

## 3.1 EMPIRICAL ANALYSIS OF OFFSET MISALIGNMENT

In this section, we conduct an empirical experiment on public datasets to evaluate the degree of alignment between image and text offsets. For each dataset, we randomly sample two classes as a pair of concept $(\alpha, \beta)$. Then, the images within each class are used alongside the related textual description (*e.g.,* label) of each class to measure offset misalignment $\mathcal{M}(\alpha, \beta)$ in a similar approach to directional loss:

$$\mathcal{M}(\alpha, \beta) = 1 - \cos(\Delta I_{\alpha\to\beta}, \Delta T_{\alpha\to\beta}),$$
$$\text{where } \Delta I_{\alpha\to\beta} = E_I\overline{(I_\beta)} - E_I\overline{(I_\alpha)}, \tag{2}$$
$$\text{and } \Delta T_{\alpha\to\beta} = E_T(T_\beta) - E_T(T_\alpha)$$

where $E_I\overline{(I_\alpha)}$ is the average embedding of all images of the class (concept) $\alpha$ in CLIP space. In addition, to measure the distance between two concepts denoted by $\mathcal{D}(\alpha, \beta)$, we use the cosine similarity between images of two classes, *i.e.*, $\mathcal{D}(\alpha, \beta) = 1 - \cos(E_I\overline{(I_\beta)}, E_I\overline{(I_\alpha)})$. We repeat this process to have $N = 5000$ pairs of concepts for each dataset. Then, we plot $\mathcal{M}(\alpha, \beta)$ against $\mathcal{D}(\alpha, \beta)$ for each pair of concepts.

**Experimental Setup.** In this experiment, we use CLIP ViT-Base/32 as the vision encoder. We use 6 large and multi-class datasets that are publicly available, including ImageNet Deng et al. (2009), Caltech-101 Fei-Fei et al. (2007), OpenImages Kuznetsova et al. (2020), MS COCO Lin et al. (2014), Visual Genome Krishna et al. (2017), and CIFAR-100 Krizhevsky et al. (2009) (details in Sec. A.1).

**Results.** Fig.2 shows the offset misalignment against the concept distance for $N = 5000$ pairs of concepts for 6 public datasets. As shown in the plots, for all datasets, apart from their different distributions and characteristics, there is a correlation between offset misalignment and concept distance. Particularly, if two concepts $\alpha$ and $\beta$ are distant, there is a higher misalignment between image offset $\Delta I_{\alpha \to \beta}$ and corresponding text offset $\Delta T_{\alpha \to \beta}$. This means that given $I_\alpha$, $T_\alpha$ and $T_\beta$, it is sub-optimal to align $\Delta I_{\alpha \to \beta}$ and $\Delta T_{\alpha \to \beta}$ to find $I_\beta$. On the other hand, if two concepts $\alpha$ and $\beta$ are closer, potentially, it is more accurate to align $\Delta I_{\alpha \to \beta}$ and $\Delta T_{\alpha \to \beta}$ to find $I_\beta$.

**Remark:** Our work is the first to find that **offset misalignment between image and text modalities in CLIP space depends on concept distance**. In what follows, we design an experiment to show that less offset misalignment leads to a better generative adaptation with directional loss.

## 3.2 IMPACT OF OFFSET MISALIGNMENT ON GENERATIVE MODEL ADAPTATION

In the previous section, we performed an empirical study that revealed the offset misalignment for natural data. In this section, we take a step further and investigate the effect of this misalignment on the generative model adaptation from a source domain (concept) $\mathcal{S}$ to a target domain (concept) $\mathcal{T}$. Specifically, following zero-shot generative domain adaptation setup Gal et al. (2022), for source domain $\mathcal{S}$, we assume a pre-trained generator $G_\mathcal{S}$ and a text description $T_\mathcal{S}$ is available. However, for the target domain, only text description $T_\mathcal{T}$ is available. To simulate different degrees of misalignment between source and target, we augment target text to get a set of text descriptions $\{T_{\mathcal{T}_i}\}$. Then, we perform zero-shot adaptation using the directional loss (Eq. 1) from the source domain $\mathcal{S}$ to each of these target text $\mathcal{T}_{\mathcal{T}_i}$ and measure the generation performance of the adapted generator.

**Experimental Setup.** For this experiment, we perform adaptation on Human → Baby and Dog → Cat. We use StyleGAN2-ADA Karras et al. (2020a) pre-trained on FFHQ Karras et al. (2019) and AFHQ-Dog Choi et al. (2020) as the pre-trained model. We fix the source text $T_\mathcal{S}$ and augment the target text $T_\mathcal{T}$ by sampling handcrafted prompts from the CLIP ImageNet template (INt)[1] which leads to increasing the misalignment (see Supp. Sec. A.2 for details). Then, we follow exactly the same hyperparameters as NADA (see Supp. Sec. A.3) to adapt the source generator to different target text $\mathcal{T}_i$. We use FID to measure the performance of the adapted generator against offset misalignment.

**Our results** in Fig. 3 demonstrates that in general, **increasing the offset misalignment degrades the performance of the zero-shot generative adaption with directional loss**. Motivated by this finding, in the next section, we propose an approach to iteratively refine the adaptation direction.

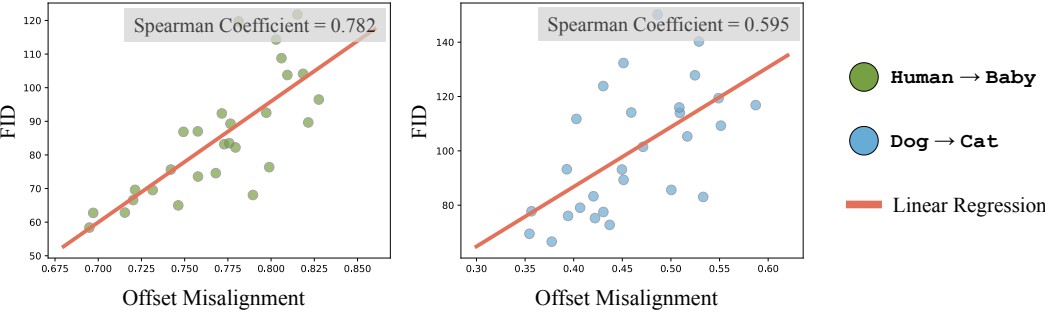

Figure 3: **Impact of offset misalignment on zero-shot generative model adaptation with directional loss:** For each of the two setups, we fix the source domain and augment the description of the target domain to achieve various degrees of misalignment between image offset and text offset. Then, we perform the adaptation using directional loss in Eq. 1 for each setup. Results show that adaptation performance degrades by increasing the offset misalignment.

---

[1] https://github.com/openai/CLIP/blob/main/notebooks/Prompt_Engineering_for_ImageNet.ipynb

---

**Algorithm 1:** Zero-Shot Learning using Adaptation with Iterative Refinement (AIR)

---

**Require:** Pre-trained generator $G_\mathcal{S}$, textual descriptions $T_\mathcal{S}$ and $T_\mathcal{T}$, $t_{adapt}$, $t_{thresh}$, $t_{int}$, learning rate $\alpha$, CLIP image and text encoder $E_I$ and $E_T$

**Output:** Trained generator $G_t$ to produce high-quality target domain images

1   Initialize $G_t$ by weights of $G_\mathcal{S}$ and freeze weights of $G_\mathcal{S}$, $i = 0$, $\mathcal{L}_{adaptive} = 0$

2   $\Delta T_{\mathcal{S} \to \mathcal{T}} = E_T(T_\mathcal{T}) - E_T(T_\mathcal{S})$

3   **for** $t = 0$; $t{+}{+}$; $t < t_{adapt}$ **do**

4      $\Delta I_{\mathcal{S} \to t} = E_I(G_t(w)) - E_I(G_\mathcal{S}(w))$

5      $\mathcal{L}_{direction} = 1 - \cos(\Delta I_{\mathcal{S} \to t}, \Delta T_{\mathcal{S} \to \mathcal{T}})$

6      **if** $t \% t_{int} = 0$ **then**

7         $i{+}{+}$

8         $G_{\mathcal{A}_i} = G_t$

9         $P_{\mathcal{A}_i} = $ Prompt-Learning $(G_{\mathcal{A}_i}, G_{\mathcal{A}_{i-1}}, P_{\mathcal{A}_{i-1}})$ /* refer to Algorithm 2 for details */

10     **end**

11     **if** $t > t_{thresh}$ **then**

12        $\Delta I_{\mathcal{A}_i \to t} = E_I(G_t(w)) - E_I(G_{\mathcal{A}_i}(w))$ /* if $G_t = G_{\mathcal{A}_i}$, add perturbation to $G_t(w)$ */

13        $\Delta T_{\mathcal{A}_i \to \mathcal{T}} = E_T(T_\mathcal{T}) - E_T(P_{\mathcal{A}_i})$

14        $\mathcal{L}_{adaptive} = 1 - \cos(\Delta I_{\mathcal{A}_i \to t}, \Delta T_{\mathcal{A}_i \to \mathcal{T}})$

15     **end**

16     $\mathcal{L} = \mathcal{L}_{direction} + \mathcal{L}_{adaptive}$

17     Update $G_t \leftarrow G_t - \alpha \nabla_{G_t} \mathcal{L}$

18   **end**

---

## 4   METHODOLOGY: ADAPTATION WITH ITERATIVE REFINEMENT

Our analysis in Sec. 3 suggests that for the closer concepts, there is less offset misalignment in CLIP space, resulting in a more accurate directional loss (Eq. 1) for adaptation. Here, we leverage this property to enhance the zero-shot generative model adaptation with directional loss.

Specifically, even though the concept distance between source $\mathcal{S}$ and target $\mathcal{T}$ is fixed, our intuition is: *'after limited iterations of adaptation using directional loss, the encoded concept in the adapted generator is already closer to the target domain than the encoded concept in source generator'*. For example, when adapting a generator pre-trained on Photo to the target domain Painting, after limited iterations, the adapted generator already encodes some knowledge related to the "Painting" domain, while this knowledge does not exist in the pre-trained generator.

Following this intuition, we use the adapted generator as the new anchor (denoted by $G_\mathcal{A}$), and compute the directional loss from this anchor point to the target. We update this anchor point iteratively during adaptation, as we move closer to the target domain. Because of the smaller concept distance, we believe the directional loss computed based on $G_\mathcal{A}$ can provide better guidance and rectify the adaptation direction solely computed based on $G_\mathcal{S}$. The major challenge of using $G_\mathcal{A}$ within directional loss is that related text prompt $P_\mathcal{A}$ that describes this concept is unknown. In what follows, first, we discuss the details of the proposed **Adaptation with Iterative Refinement (AIR)** in Sec. 4.1. Then, to address the challenge of unknown $P_\mathcal{A}$ within the directional loss of AIR, we discuss the proposed prompt learning approach in Sec. 4.2.

### 4.1   ADAPTATION WITH ITERATIVE REFINEMENT (AIR)

In our proposed approach, first, we adapt the generator to the target domain for $t_{thresh}$ iterations using directional loss in Eq. 1 to make sure the adapted generator has moved closer to the target domain. Then, in each $t_{int}$ interval of adaptation, we sample the adapted generator as the new anchor point. We denote $i^{th}$ sampled anchor by $G_{\mathcal{A}_i}$. To reduce offset misalignment and provide more accurate direction, we use the anchor point $\mathcal{A}_i$ instead of source point $\mathcal{S}$ for computing the directional loss. The image offset with anchor point $\mathcal{A}_i$ is computed based on the sampled generator $G_{\mathcal{A}_i}$, and the trainable generator $G_t$: $\Delta I_{\mathcal{A}_i \to t} = E_I(G_t(w)) - E_I(G_{\mathcal{A}_i}(w))$. Assuming that the anchor point is described by the prompt $P_{\mathcal{A}_i}$ in the text domain (details of acquiring $P_{\mathcal{A}_i}$ will be discussed in Sec. 4.2), the text offset with anchor point is calculated as follows: $\Delta T_{\mathcal{A}_i \to \mathcal{T}} = E_T(T_\mathcal{T}) - E_T(P_{\mathcal{A}_i})$.

---

**Algorithm 2:** Proposed Prompt Learning

---

**Require:** Current and previous anchor generators $G_{\mathcal{A}_i}$ and $G_{\mathcal{A}_{i-1}}$, learned text prompt for previous anchor $P_{\mathcal{A}_{i-1}}$, learning rate $\beta$, CLIP image and text encoder $E_I$ and $E_T$

**Output:** Prompt vector $P_{\mathcal{A}_i}$ to represent current anchor.

1   $\Delta I_{\mathcal{A}_{i-1} \to \mathcal{A}_i} = E_I(G_{\mathcal{A}_i}(w)) - E_I(G_{\mathcal{A}_{i-1}}(w))$

2   **for** $k = 0; k{+}{+}; k < k_{iter}$ **do**

3     $\Delta P_{\mathcal{A}_{i-1} \to \mathcal{A}_i} = E_T(P_{\mathcal{A}_i}) - E_T(P_{\mathcal{A}_{i-1}}).$

4     $\mathcal{L}_{align} = 1 - \cos(\Delta I_{\mathcal{A}_{i-1} \to \mathcal{A}_i}, \Delta P_{\mathcal{A}_{i-1} \to \mathcal{A}_i})$

5     Update $P_{\mathcal{A}_i} \leftarrow P_{\mathcal{A}_i} - \beta \nabla_{P_{\mathcal{A}_i}} \mathcal{L}_{align}$

6   **end**

---

Finally, the adaptive loss $\mathcal{L}_{adaptive}$ is computed by aligning the image and text offsets from anchor point $\mathcal{A}_i$ to target $\mathcal{T}$:

$$\mathcal{L}_{adaptive} = 1 - \cos(\Delta I_{\mathcal{A}_i \to t}, \Delta T_{\mathcal{A}_i \to \mathcal{T}}) \tag{3}$$

We empirically find that adding this adaptive loss to $\mathcal{L}_{direction}$ results in a more stable adaptation. The proposed AIR scheme is summarized in Alg.1.

### 4.2 ALIGNING PROMPT TO IMAGES

Here, we explain the details of the proposed method for learning the text prompt $P_{\mathcal{A}_i}$ that describes the $i^{th}$ anchor point $\mathcal{A}_i$ in text domain. Inspired by Zhou et al. (2022b), we define the prompt $P_{\mathcal{A}_i} \in \mathbb{R}^{(M+1) \times d}$ as combination of $M$ learnable tokens $[V]_j^i \in \mathbb{R}^d$, and a label token $Y_{\mathcal{A}_i} \in \mathbb{R}^d$:

$$P_{\mathcal{A}_i} = [V]_1^i [V]_2^i \dots [V]_M^i [Y_{\mathcal{A}_i}] \tag{4}$$

Early approaches of prompt learning directly learn the learnable tokens $[V]_j^i$ from related images Zhou et al. (2022b;a). However, recently, ITI-GEN Zhang et al. (2023) shows that learning from the offsets is more efficient for capturing the specific attribute of interest.

**Remark:** Similar to NADA, ITI-GEN also uses the mechanism of the aligning offsets for learning target prompts which could be susceptible to offset misalignment. However, there are two major differences between them: i) ITI-GEN aligns $\Delta P$ to the target $\Delta I$ which could provide a better supervisory signal, ii) the number of trainable parameters in ITI-GEN is significantly lower (2K in ITI-GEN compared to 2.7 millions in NADA). In what follows, we propose two design choices to make learning $\Delta P$ less susceptible to the offset misalignment issue.

**Design Choices.** To mitigate the possible issue of offset misalignment, we propose to decrease the distance between concept pairs and use a regulirzer token during prompt learning. More specifically:

- Given that offset misalignment is less for closer concepts, we use the previous anchor point $\mathcal{A}_{i-1}$ as the source to learn the prompt for the $i^{th}$ anchor $\mathcal{A}_i$. Since consecutive anchor points are close together, the directional loss is more accurate.

- We use the interpolation between tokenized source and target descriptions as anchor label, *i.e.*, $Y_{\mathcal{A}_i} = (1 - p_i)Y_{\mathcal{S}} + p_i Y_{\mathcal{T}}$, with $p_i$ denoting the proportion of the training progress until anchor point $\mathcal{A}_i$. The label token acts like a regularizer during prompt learning.

We empirically find that using these two design choices results in better adaptation with our AIR mechanism compared to learning the prompts directly from generated images by $G_{\mathcal{A}_i}$.

Therefore, the image offset is calculated between the current and previous anchors: $\Delta I_{\mathcal{A}_{i-1} \to \mathcal{A}_i} = E_I(G_{\mathcal{A}_i}(w)) - E_I(G_{\mathcal{A}_{i-1}}(w))$. Similarly, the text prompt offset is calculated as follows: $\Delta P_{\mathcal{A}_{i-1} \to \mathcal{A}_i} = E_T(P_{\mathcal{A}_i}) - E_T(P_{\mathcal{A}_{i-1}})$. Note that the only trainable parameter is the unknown prompt $P_{\mathcal{A}_i}$ which is learned by aligning image and prompt offsets:

$$\mathcal{L}_{align} = 1 - \cos(\Delta I_{\mathcal{A}_{i-1} \to \mathcal{A}_i}, \Delta P_{\mathcal{A}_{i-1} \to \mathcal{A}_i}) \tag{5}$$

The proposed prompt learning approach is summarized in Alg. 2. We remark that $P_{\mathcal{A}_i}$ is the tokenized text prompt before the CLIP text encoder, and for simplicity, we slightly abuse the notation and use $E_T(P_{\mathcal{A}_i})$ to show CLIP text embedding for anchor $\mathcal{A}_i$.

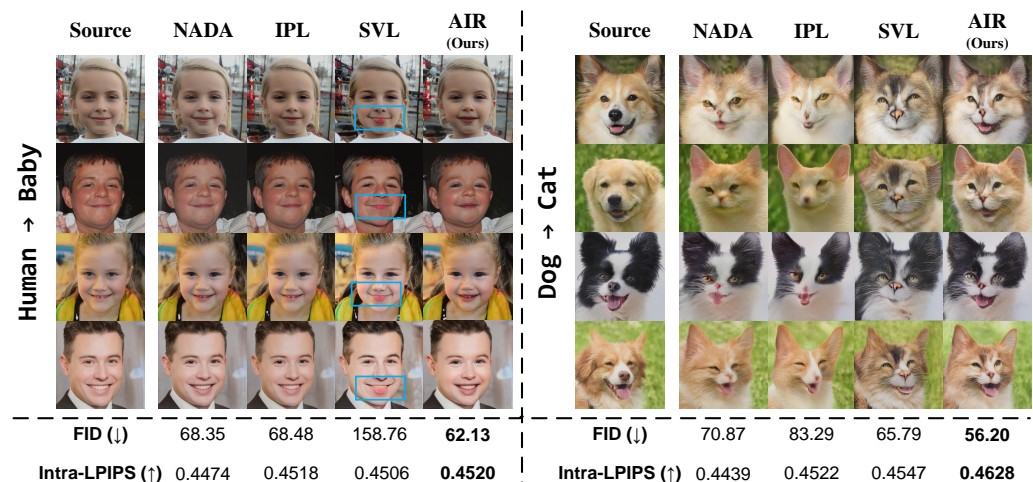

Figure 4: **Qualitative and quantitative comparison results:** proposed AIR achieves SOTA performance indicated by lower FID (better quality) and higher Intra-LPIPS (higher diversity). Here, we use StyleGAN2 as the pre-trained generator (qualitative results for diffusion model in Supp).

## 5 EXPERIMENTS

In this section, first, we discuss the details of our experimental setup. Then, we compare our proposed AIR method with SOTA approaches both quantitatively and qualitatively. We are the first work in the literature that studies zero-shot adaptation of the diffusion models. Finally, we conduct an ablation study on the design of our prompt learning strategy.

### 5.1 EXPERIMENTAL SETUP

**Generative Models.** In this work, we implement zero-shot generative model adaptation for both GANs and diffusion models. Note that **we perform the first zero-shot adaptation on the diffusion model**. The implementation details for each type of model is as follows:

- **Zero-Shot Adaptation of GANs.** In this setup, we follow the previous works Gal et al. (2022); Guo et al. (2023); Jeon et al. (2023) settings to adapt StyleGAN2-ADA Karras et al. (2020a) pre-trained on FFHQ Karras et al. (2019) and AFHQ-Dog Choi et al. (2020) to various target domains.

- **Zero-Shot Adaptation of Diffusion Models.** In this setup, we use Guided Diffusion Dhariwal & Nichol (2021) pre-trained on FFHQ and AFHQ-Dog from P2-Weighting Choi et al. (2022) as our source generator. To speedup training, we use DPM-Solver Lu et al. (2022) to generate images in 10 steps. To prevent overfitting, instead of fully fine-tuning the pre-trained model, we fine-tune it with LoRA Hu et al. (2022).

During the adaptation of both generators, we utilize the pre-trained ViT-Base/32 Dosovitskiy et al. (2021) as the vision encoder for CLIP Radford et al. (2021). The details of the hyperparameters used during the adaptation can be found in Supp. Sec. A.4.

**Evaluation Metrics.** A well-trained image generator is defined by its ability to produce high-quality and diverse images from target distribution. Following the zero-shot works in literature, in this work we conduct both visual inspections for qualitative evaluations and quantitative evaluations using the following metrics:

- **FID.** For target domains with large and publicly available datasets (e.g., Baby and Cat), we follow previous work Jeon et al. (2023) to use these datasets as target distribution. Then, we generate 5000 samples for each target domain Zhao et al. (2022a; 2023), and use FID to evaluate the generated images' quality and diversity.

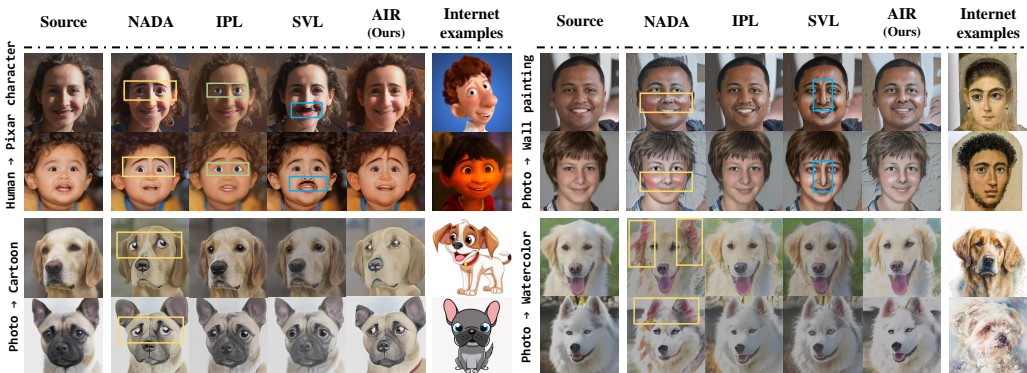

Figure 5: **Qualitative comparison results:** proposed method does not suffer from artifacts, and adapts better to the style of the target domains. StyleGAN2 is used as the pre-trained generator.

- **CLIP Distance.** The public data is scarce for other target domains, *e.g.,* Pixar. For these domains, following Guo et al. (2023) we collect a set of images using a simple query and crawling process. Then, we use the CLIP Distance Gal et al. (2023) which is defined as the cosine distance between the clip embeddings of the collected images and the generated images to measure the similarity of the generated images to the target domain.

- **Intra-LPIPS.** To measure the diversity of the generated images, we use Intra-LPIPS metric Ojha et al. (2021) which first assigns generated images to one of $K$ clusters, then averages pair-wise distance within the cluster members and reports the average value over $K$ clusters. In zero-shot setup, since there are no training images, we follow Gal et al. (2022); Jeon et al. (2023) to cluster around generated images using $K$-Medoids Kaufman & Rousseeuw (2009), with $K = 10$.

- **User Study.** We also conduct a user study to compare the quality and the diversity of the generated images with different schemes based on human feedback. The questionnaire is performed using the generated images by different schemes including NADA, IPL, SVL, and our proposed AIR. It includes 12 questions for quality evaluation and 4 questions for diversity assessment. Finally, we report the percentage of the user preference for each method and for both quality and diversity metrics.

More details about the evaluation can be found in Supp. Sec. A.5.

## 5.2 EXPERIMENTAL RESULTS

**Qualitative results.** We report qualitative results of zero-shot GAN adaptation across a wide range of target domains and compare with SOTA methods Gal et al. (2022), Guo et al. (2023), Jeon et al. (2023) as shown in Fig.1, 4, and 5. The results of NADA show the adaptation often introduces undesirable changes in features, e.g., eyes shifting in Photo → Cartoon and red cheeks in Photo → Wall painting. For IPL and SVL, the adaptations are incomplete, resulting in images that retain too much of the source domain features, especially for adaptations that require drastic feature change, such as Human → Werewolf and Photo → Sketch. Our proposed method can fully adapt to the target domain without generating undesired features. We also show the qualitative results of zero-shot diffusion model adaptation in Fig. 6, and additional results for GAN adaptation in Fig. 11.

**Quantitative results.** We report FID, Intra-LPIPS, and CLIP Distance to quantify the performance of zero-shot adaptation of both GAN and diffusion model. As shown in Fig.1 and Tab. 1, 2, IPL and SVL perform better for most scenarios in terms of diversity compared to NADA, but the quality is degraded. We emphasize that our method AIR significantly outperforms SOTA methods in both quality and diversity by performing more accurate adaptation in CLIP space. Our user study results in Tab. 3 further confirm the advancement of our method in both quality and diversity.

**Ablation Study.** We conduct ablation studies to verify the effectiveness of our prompt learning design. We compare three different schemes to learn prompts and fix all other settings to perform ZSGM with iterative refinement: i) $\mathcal{I} \rightarrow \mathcal{T}$: We follow IPL to learn a latent mapper that directly produces prompt descriptions from each image. ii) $\mathcal{S} \rightarrow \mathcal{A}_i$: We learn the

Table 1: Quantitative evaluation of zero-shot GAN adaptation. Best results are **bold**.

| Pre-trained Dataset | Adaptation | CLIP Distance ($\downarrow$) | | | | Intra-LPIPS ($\uparrow$) | | | |
| --- | --- | --- | --- | --- | --- | --- | --- | --- | --- |
| | | NADA | IPL | SVL | AIR | NADA | IPL | SVL | AIR |
| FFHQ | Human $\rightarrow$ Baby | 0.3327 | 0.3562 | 0.3838 | **0.3325** | 0.4474 | 0.4518 | 0.4506 | **0.4520** |
| | Human $\rightarrow$ Werewolf | 0.3175 | 0.2819 | 0.3868 | **0.2125** | 0.4114 | 0.4387 | 0.4316 | **0.4410** |
| | Human $\rightarrow$ Pixar | 0.2335 | 0.2343 | 0.4224 | **0.2213** | 0.4759 | 0.4488 | 0.4618 | 0.4717 |
| | Photo $\rightarrow$ Sketch | 0.3739 | 0.3955 | 0.4092 | **0.3469** | 0.3870 | 0.4292 | 0.4476 | **0.4493** |
| | Photo $\rightarrow$ Wall painting | 0.4382 | 0.4898 | 0.4952 | **0.4306** | 0.4217 | 0.4320 | 0.4332 | **0.4381** |
| AFHQ-Dog | Dog $\rightarrow$ Cat | 0.1493 | 0.1530 | 0.1644 | **0.1320** | 0.4439 | 0.4522 | 0.4547 | **0.4628** |
| | Photo $\rightarrow$ Cartoon | 0.2433 | 0.2419 | 0.2543 | **0.2258** | 0.4356 | 0.4413 | 0.4400 | **0.4427** |
| | Photo $\rightarrow$ Walltercolor | 0.1535 | 0.1711 | 0.1646 | **0.1507** | 0.4639 | **0.4703** | 0.4622 | 0.4665 |

Table 2: Quantitative evaluation of zero-shot diffusion model adaptation.

| Pre-trained Dataset | Adaptation | FID ($\downarrow$) | | CLIP Distance ($\downarrow$) | | Intra-LPIPS ($\uparrow$) | |
| --- | --- | --- | --- | --- | --- | --- | --- |
| | | NADA | AIR | NADA | AIR | NADA | AIR |
| FFHQ | Human $\rightarrow$ Baby | 65.54 | **58.05** | 0.2598 | **0.2162** | 0.5700 | **0.5779** |
| | Photo $\rightarrow$ Sketch | - | - | 0.4405 | **0.3576** | **0.4868** | 0.4860 |
| AFHQ-Dog | Dog $\rightarrow$ Cat | 85.02 | **77.61** | 0.1406 | **0.1402** | 0.5423 | **0.5445** |
| | Photo $\rightarrow$ Cartoon | - | - | 0.2544 | **0.2472** | 0.5574 | **0.5603** |

Table 3: Results of our user study (%) experiments.

| Evaluation | Quality | Diversity |
| --- | --- | --- |
| NADA | 29.55 | 22.73 |
| IPL | 3.03 | 15.91 |
| SVL | 8.33 | 4.54 |
| AIR | **59.09** | **56.82** |

Table 4: Ablation study on prompt learning scheme.

| Methods | Human $\rightarrow$ Baby | | Dog $\rightarrow$ Cat | |
| --- | --- | --- | --- | --- |
| | FID ($\downarrow$) | Intra-LPIPS ($\uparrow$) | FID ($\downarrow$) | Intra-LPIPS ($\uparrow$) |
| NADA | 68.35 | 0.4474 | 70.87 | 0.4439 |
| $\mathcal{I} \rightarrow \mathcal{T}$ | 98.35 | 0.4308 | 104.59 | 0.4452 |
| $\mathcal{S} \rightarrow \mathcal{A}_i$ | 64.39 | 0.4503 | 61.75 | **0.4630** |
| $\mathcal{A}_{i-1} \rightarrow \mathcal{A}_i$ | **62.13** | **0.4520** | **56.20** | 0.4628 |

prompt by capturing the semantic difference between $\mathcal{S}$ and $\mathcal{A}$ with directional loss, denoted as $\mathcal{L}_{align}^{\mathcal{S}} = 1 - \cos(\Delta I_{\mathcal{S} \rightarrow \mathcal{A}_i}, \Delta P_{\mathcal{S} \rightarrow \mathcal{A}_i})$. iii) $\mathcal{A}_{i-1} \rightarrow \mathcal{A}_i$: Our proposed prompt learning scheme, which captures the semantic difference between consecutive anchors $\mathcal{A}_{i-1}$ and $\mathcal{A}$ with directional loss as denoted in Eq. 5. Tab. 4 shows that learning prompts directly from images cannot produce accurate text descriptions for anchor domains. $\mathcal{A}_{i-1} \rightarrow \mathcal{A}_i$ suffer less from offset misalignment compared to $\mathcal{S} \rightarrow \mathcal{A}_i$, therefore the learned prompt is more accurate and results in better zero-shot adaptations.

**Additional Experiments.** We conduct additional experiments to demonstrate the well-behaved latent space of the pre-trained generator is preserved in our method. We report results in Supp due to lack of space. Specifically, we perform latent space interpolation (Sec. C), cross-model interpolation (Sec. D), and cross-domain image manipulation with both GAN and diffusion model (Sec. E).

## 6 CONCLUSION

Previous methods in ZSGM assume that image offset and text offset are perfectly aligned in CLIP embedding space. In this paper, inspired by the studies in analogical reasoning of NLP, we conduct an empirical study to analyze the misalignment between image offset and text offset in CLIP space. Our analysis reveals that there is offset misalignment in CLIP space which correlated with concept distances. Building on this insight, we propose AIR, a new approach that iteratively samples anchor points closer to the target and mitigates offset misalignment issues. Extentsive experimental results on both GAN and diffusion models shows that the proposed AIR achieves SOTA performance across various setups.

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

SUPPLEMENTAL MATERIAL

Please find the following anonymous link for code and other resources: `https://drive.google.com/drive/folders/1zJlo5URefAhpcUrSd4TsxVOCH14gEM8v?usp=sharing`

## A DETAILED EXPERIMENTAL SETTING

### A.1 DETAILS OF EMPIRICAL ANALYSIS

For datasets with a single class label for each image, such as ImageNet, Caltech-101, and CIFAR-100, we use the original images from the dataset. For datasets with multiple objects in an image, such as OpenImages, MS COCO, and Visual Genome, to better align with the setting in NADA, we extract the objects using bounding boxes and classify them into their labeled classes.

For a certain concept $\alpha$, we use the images of the class as $I_\alpha$ and the corresponding class label with INt as $T_\alpha$.

### A.2 DETAILS OF IMPACT OF OFFSET MISALIGNMENT

We randomly sample prompt template from INt, and perform zero-shot adaptation with NADA as shown in Fig. 3. We list the details of the sampled prompts and their offset misalignment $\mathcal{M}$ as well as the adaptation quality (measured by FID) in Tab. 5.

Table 5: Prompt templates used in Sec. 3.2.

| Prompts | Human→Baby | | Dog→Cat | |
| --- | --- | --- | --- | --- |
| | Offset Misalignment | FID | Offset Misalignment | FID |
| A bad photo of a { }. | 0.6971 | 62.76 | 0.3545 | 69.47 |
| A sculpture of a { }. | 0.7895 | 68.08 | 0.4713 | 101.49 |
| A photo of the hard to see { }. | 0.7989 | 76.36 | 0.4219 | 75.24 |
| A low resolution photo of the { }. | 0.7729 | 83.18 | 0.3942 | 76.06 |
| A rendering of a { }. | 0.7577 | 73.56 | 0.4028 | 111.74 |
| Graffiti of a { }. | 0.7715 | 92.34 | 0.5332 | 83.03 |
| A bad photo of the { }. | 0.7202 | 66.58 | 0.3774 | 66.58 |
| A cropped photo of the { }. | 0.8215 | 89.66 | 0.4512 | 132.33 |
| A tattoo of a { }. | 0.8060 | 108.78 | 0.5490 | 119.40 |
| The embroidered { }. | 0.8185 | 104.13 | 0.5514 | 109.27 |
| A photo of a hard to see { }. | 0.7680 | 74.58 | 0.4066 | 79.07 |
| A bright photo of a { }. | 0.7315 | 69.54 | 0.4305 | 77.50 |
| A dark photo of the { }. | 0.7758 | 83.50 | 0.4592 | 114.12 |
| A drawing of a { }. | 0.7765 | 89.28 | 0.4304 | 123.84 |
| A photo of my { }. | 0.6949 | 58.39 | 0.3566 | 77.76 |
| The plastic { }. | 0.7812 | 119.73 | 0.5092 | 113.99 |
| A photo of the cool { }. | 0.8094 | 103.78 | 0.4496 | 93.12 |
| A close-up photo of a { }. | 0.7213 | 69.61 | 0.4370 | 72.75 |
| A black and white photo of the { }. | 0.7463 | 64.99 | 0.5288 | 140.25 |
| A painting of the { }. | 0.8152 | 121.74 | 0.4862 | 150.15 |
| A painting of a { }. | 0.7576 | 87.01 | 0.4513 | 89.32 |
| A pixelated photo of the { }. | 0.7154 | 62.85 | 0.5168 | 105.32 |
| A sculpture of the { }. | 0.7794 | 82.22 | 0.5086 | 115.97 |
| A bright photo of the { }. | 0.8029 | 114.31 | 0.4203 | 83.28 |
| A cropped photo of a { }. | 0.7493 | 86.87 | 0.3929 | 93.22 |
| A plastic { }. | 0.7420 | 75.65 | 0.5247 | 127.82 |
| A photo of the dirty { }. | 0.8276 | 96.47 | 0.5004 | 85.62 |
| A jpeg corrupted photo of a { }. | 0.7972 | 92.56 | 0.5872 | 88.73 |

A.3    Hyperparameters of impact of offset misalignment

For the hyperparameter choices in Sec. 3.2, we strictly follow the settings in NADA except that only the ViT-B/32 is used as vision encoder. The details of hyperparameters are shown in Tab. 6.

Table 6: Hyperparameters choices of NADA in Sec. 3.2.

| Source | Target | Prompt template | Iterations | Adaptive k | Mixing |
|--------|--------|-----------------|------------|------------|--------|
| Human  | Baby   | INt             | 300        | 18         | 0.0    |
| Dog    | Cat    | INt             | 2000       | 3          | 0.0    |

A.4    Hyperparameters of zero-shot adaptation

In Alg. 1, for both GAN and diffusion model adaptation the batch size is set to 2. Adaptation iteration $t_{adapt}$ is set to 300 for texture-based changes such as Photo→Sketch, and 2,000 for animal changes like Dog→Cat. We set $t_{thresh} = 0.6t_{adapt}$ to ensure there are some target domain concept encoded in $G_t$, and $t_{int} = 0.1t_{adapt}$ to facilitate a stable and efficient training.

In Alg. 2, we generate 1,000 pairs of source and anchor images with the same batch of $w$ for each update. The number of prompt vectors $m$ is set to 4, and is initialized by "A photo of a". Each of the prompt learning sessions requires $k_{iter} = 200$ iterations.

For all experiments, we use an ADAM optimizer with a learning rate of 0.002. We conduct all the experiments on a single NVIDIA RTX 6000 Ada GPU. The training time is comparable to NADA as prompt learning in Alg. 2 only requires ∼20 seconds in our environment.

A.5    Evaluataion details

We did our best to follow existing zero-shot works in evaluation setup and further improve on them. Specifically, following previous works Gal et al. (2022); Guo et al. (2023); Jeon et al. (2023), we have conducted comparisons on both public datasets and images collected from internet. For evaluation on the public datasets, we have used FFHQ-Baby Ojha et al. (2021) (for target domain Baby), and AFHQ-Cat Choi et al. (2020) (for target domain Cat). We report FID and Intra-LPIPS on these two datasets for different approaches. We remark that similarly NADA reports Intra-LPIPS on AFHQ-Cat, and SVL reports both FID and Intra-LPIPS on AFHQ-Cat.

For evaluation of collected images from the internet, we follow IPL's idea to collect internet images as reference. However, since IPL did not make the collected images publicly available, we had to repeat this practice and collect the images.

In addition, we believe the included visual results in all cases, can help in transparency and reflecting the superior performance of our proposed method in terms of adaptation quality.

B    Zero-shot Diffusion model adaptation

In this section, we show the results of zero-shot diffusion model adaptation. As illustrated in Fig. 6, the generated images of baseline NADA suffer from mode collapse issue, e.g., wrinkle in Human → Baby, missing left eye in Photo → Cartoon. Our AIR method can generate images that encode more target domain information while preserving the diversity. Note that both methods struggles to generate cat images for Dog → Cat. This could be because diffusion models learn to model the exact data distribution, therefore it is inherently tied to the learned data distribution and struggle to generate images from substantially different domains like cats in a zero-shot setting.

C    Latent Space Interpolation

Building on prior research, we demonstrate that the target domain generators refined through our method retain a smooth latent space property. As illustrated in Fig.7, each row features a series of images from the same target domain. The left-most and right-most images in each row, labeled as

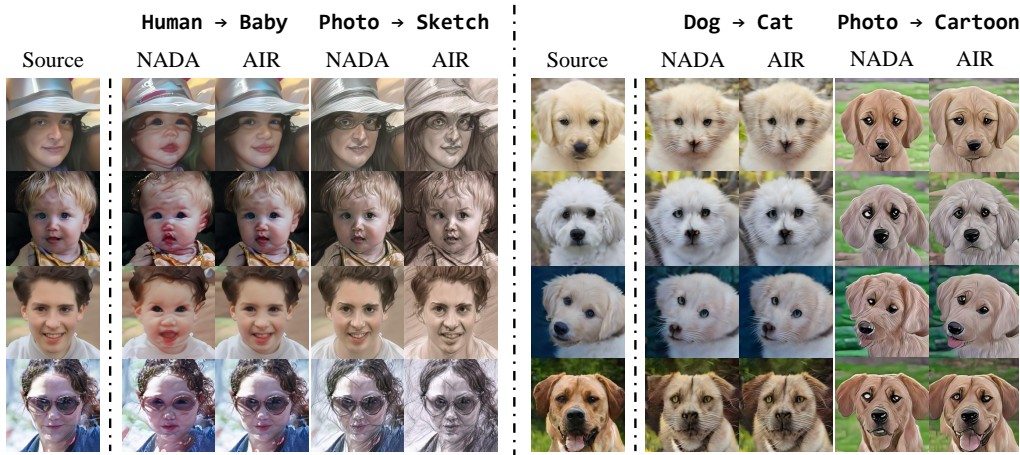

Figure 6: Qualitative results of zero-shot diffusion model adaptation.

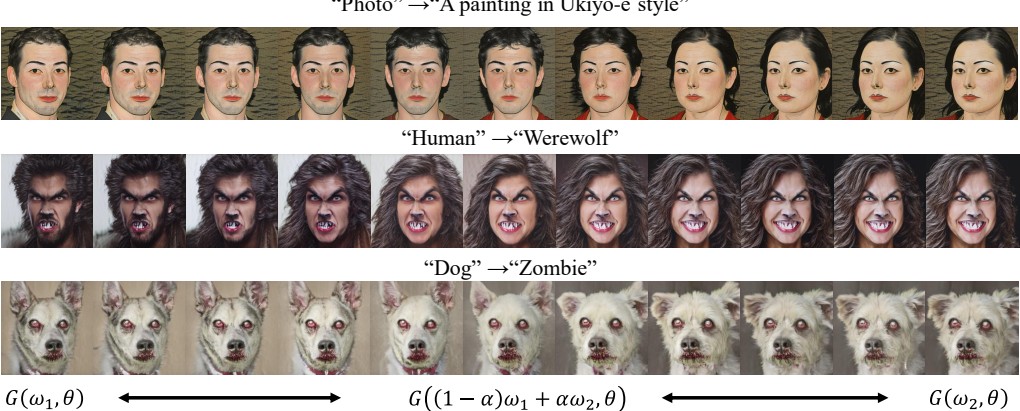

Figure 7: Latent space interpolation. For each row, the left-most column and right-most column are respectively two images synthesized with two different latent codes. The remaining columns refer to images synthesized with interpolated latent codes.

$G_t(w_1)$ and $G_t(w_2)$ respectively, are generated using distinct latent codes $w_1$ and $w_2$. Latent space interpolation between these codes produces an image $G_t((1-\alpha)w_1 + \alpha w_2)$, where $\alpha$ varies from 0 to 1. The visual results show that our method has good robustness and generalization ability. The various target domain spaces obtained by our method are consistently smooth.

## D  CROSS-MODEL INTERPOLATION

In addition to demonstrating latent space interpolation, we also explore the model's weight smoothness across various domains. Specifically, we perform linear interpolation in the weight space between $G(\cdot, \theta_s)$ and $G(\cdot, \theta_{t_1})$, or between $G(\cdot, \theta_{t_1})$ and $G(\cdot, \theta_{t_2})$. Here, $G(\cdot, \theta_s)$ represents the source domain generator, while $G(\cdot, \theta_{t_1})$ and $G(\cdot, \theta_{t_2})$ are generators adapted to two different target domains. Given a latent code $w$, we produce images via an interpolated model, $G(w, (1-\alpha)\theta_1 + \alpha\theta_2)$, where $\alpha$ ranges from 0 to 1. As illustarted in Fig.8, our approach effectively supports smooth cross-model interpolation, whether transitioning from a source to a target domain or between different target domains.

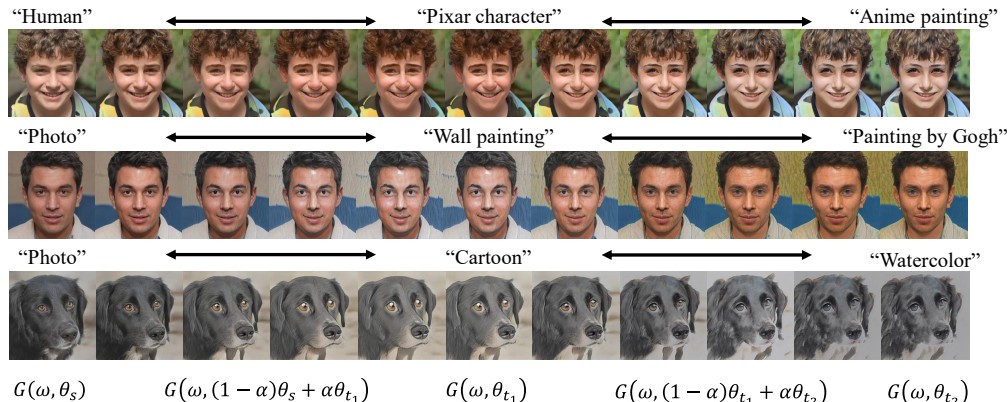

$$G(\omega, \theta_s) \qquad G(\omega, (1-\alpha)\theta_s + \alpha\theta_{t_1}) \qquad G(\omega, \theta_{t_1}) \qquad G(\omega, (1-\alpha)\theta_{t_1} + \alpha\theta_{t_2}) \qquad G(\omega, \theta_{t_2})$$

Figure 8: Cross-model interpolation. In each row, the left-most image is generated by the source generator. The middle and the right-most images are synthesized by two different target domain generators. The other images represent cross-model interpolations between two different domains.

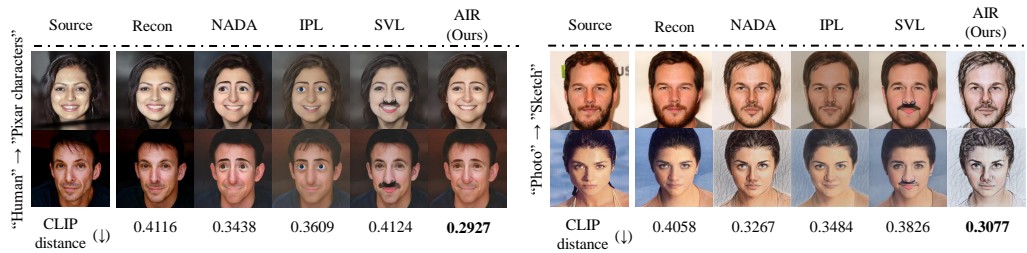

Figure 9: Image manipulation with GAN. The reference image are the same as in Fig. 1, 5.

## E  IMAGE MANIPULATION

To further demonstrate the effectiveness of our proposed method, we also conduct experiments on text-to-image manipulation. It first inverts a image to the latent code by a pre-trained inversion model and then feeds it to the trained target domain generator to get the translated target domain image.

We experiment on both GAN and diffusion model. We use RestyleAlaluf et al. (2021) with e4e encoder Tov et al. (2021) to invert a real image into the latent space $w$ for StyleGANs. For the diffusion model, we follow the setting of DiffusionCLIP Kim et al. (2022) to diffuse a real image and fintune the model to generate an image with target domain features using the diffused image.

### E.1  GAN-BASED IMAGE MANIPULATION

For GAN-based generators, we perform the experiment by utilizing the inversion model Restyle Alaluf et al. (2021) with e4e encoder Tov et al. (2021). As illustrated in Fig 9, our method qualitatively exhibits a higher fidelity of target domain features compared to previous methods. Quantitatively, our approach more closely aligns with the reference target images in CLIP space, indicating a greater semantic similarity.

### E.2  DIFFUSION-BASED IMAGE MANIPULATION

We implement based on Diffusion-CLIP Kim et al. (2022) which seamlessly integrates with the existing zero-shot adaptation methods.

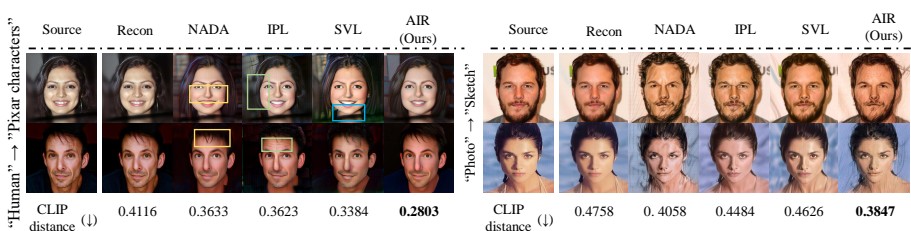

Figure 10: Diffusion model image manipulation. The reference images are the same as in Fig.1, 5.

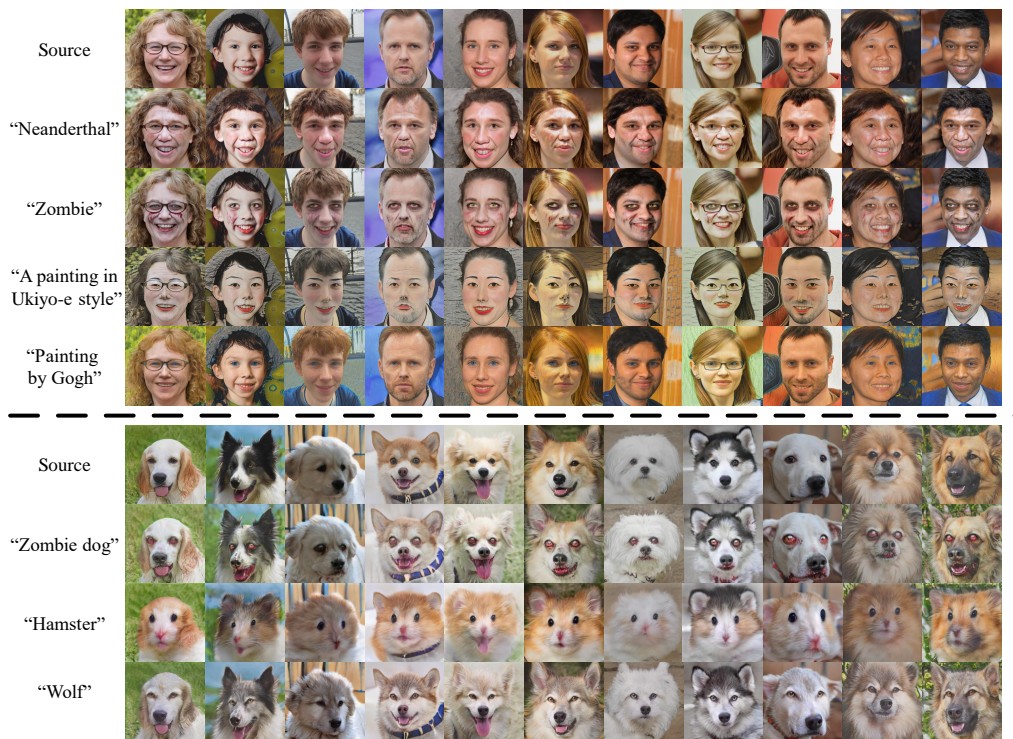

Figure 11: Additional results of zero-shot GAN adaptation with AIR.

As illustrated in Fig 10, our method qualitatively exhibits a higher fidelity of target domain feature compared to previous methods. Quantitatively, our approach more closely aligns with the reference target images in CLIP space, indicating a greater semantic similarity.

## F  ADDITIONAL VISUAL RESULTS

We present additional visual results for domain adaption and image manipulation. Specifically, Fig.11 showcases further generative model adaptation outcomes for AIR, while Fig.12 illustrates real-world image manipulation results for diffusion AIR. While domain adaptation brings more target features, image manipulation with diffusion model retains more image-specific features.

## G  RELATED WORK

**Zero-shot Generative Model Adaptation** Zero-shot generative model adaptation is the task of adapting the source domain knowledge of a well-trained generator to the target domain without

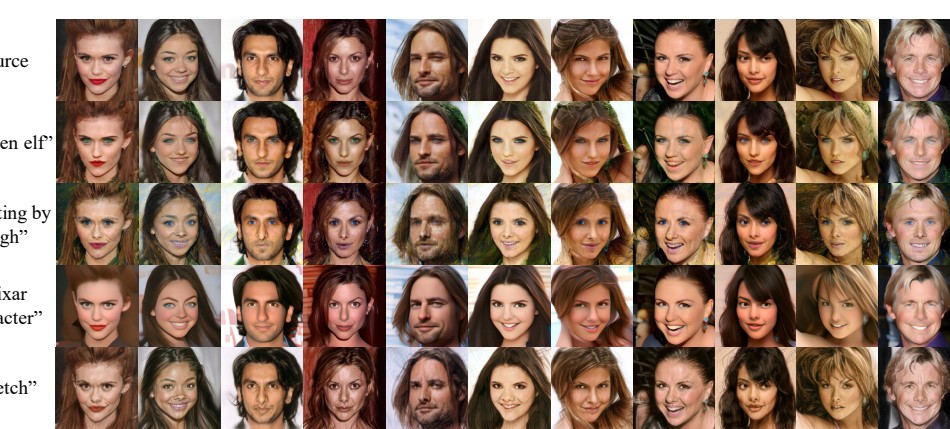

Figure 12: Additional results of image manipulation with diffusion model.

accessing any target samples. Unlike the zero-shot image editing methods Patashnik et al. (2021) Shen & Zhou (2021) where available modifications are constrained in the domain of the pre-trained generator, zero-shot generator adaptation can perform out-of-domain manipulation by directly optimizing the generator parameters. Previous works Gal et al. (2022); Guo et al. (2023); Jeon et al. (2023) utilized the cross-modal representation in CLIP Radford et al. (2021) to bypass the need for extensive data collection. Specifically, **NADA** Gal et al. (2022) first proposes to use the embedding offset of textual description in the CLIP space to describe the difference between source and target domains. By assuming the text offset and image offset are well-aligned in CLIP space, it uses the text offset as adaptation direction and optimizes the trainable generator to align image offset with text offset. **IPL** Guo et al. (2023) points out that adaptation directions in NADA for diverse image samples is computed from one pair of manually designed prompts, which will cause mode collapse, therefore they produce different adaptation directions for each sample. Similarly, **SVL** Jeon et al. (2023) use embedding statistics (mean and variance) for producing adaptation direction instead of only mean of embeddings in NADA to prevent mode collapse.

However, the adaptation direction in previous work only focuses on the source and target domains and computes once before the generator adaptation. More importantly, all these methods assume the image and text offsets in the CLIP space are well aligned. In this paper, we draw inspiration from a similar problem called analogical reasoning in NLP, and empirically discover that the alignment of image and text offset in CLIP space is correlated to the concept proximity in CLIP space. Based on this finding, we proposed a method that iteratively updates the adaptation direction, which is more aligned with the image offset and more accurate for zero-shot adaptation with directional loss.

**Analogical Reasoning** Research in NLP has shown that word representations of language models are surprisingly good at capturing semantic regularities in language Collobert & Weston (2008); Turian et al. (2010). Specifically, analogical reasoning Mikolov et al. (2013c;a;b); Levy & Goldberg (2014), utilizing the semantic regularities of word representations, aims to solve analogy tasks by using one pair of word vectors to identify the unknown member of a different pair of words, commonly via alignment of offsets, This is commonly modeled as using the vector offset between two words $a' - a$, and applying it to a new word $b$ to predict the missing word $b'$ that pair with $b$, as illustrated by the famous example of using $v(\text{``Man''})$ - $v(\text{``Woman''})$ and $v(\text{``King''})$ to identify $v(\text{"Queen"})$, where $v(\cdot)$ denotes word representation. This approach attracted a lot of attention for the vital role that analogical reasoning plays in human cognition for discovering new knowledge and understanding new concepts. It is already used in many downstream NLP tasks, such as splitting compounds Daiber et al. (2015), semantic search Cohen et al. (2015), cross-language relational search Duc et al. (2015), etc.

Importantly, previous works Levy et al. (2015); Köper et al. (2015); Vylomova et al. (2015) demonstrate that the effectiveness of analogical reasoning varies across different categories and semantic relations. More recent studies Rogers et al. (2017); Fournier et al. (2020), present a series of experiments performed with BATS dataset Gladkova et al. (2016) on various pre-trained vector space, e.g.,

| Models | License |
|---|---|
| StyleGAN2 Karras et al. (2020b) | Nvidia Source Code License |
| CLIP Radford et al. (2021) | MIT License |
| StyleGAN2-pytorch Karras et al. (2020b) | MIT License |
| e4e Tov et al. (2021) | MIT License |
| StyleGAN-NADA Gal et al. (2022) | MIT License |
| IPL Guo et al. (2023) | MIT License |
| **Datasets** | **License** |
| FFHQ [5] | CC BY-NC-SA 4.0 |
| AFHQ [1] | CC BY NC 4.0 |

Table 7: Sources and licenses of the utilized models and datasets

GloVe Pennington et al. (2014), Word2Vec Mikolov et al. (2013b), and Skip-gram Mikolov et al. (2013a), indicate that it is more effective to use $a' - a$ and $b$ to determine $b'$ when $b$ and $b'$ are close in vector space; and less so when $b$ and $b'$ are more apart.

Inspired by these studies, in this work, we perform an empirical study of offset misalignment in CLIP space and observe that for distant concepts in CLIP, image and text offset suffer from more misalignment, while closely related concepts suffer less. Based on our analysis, we proposed a method that iteratively refined the text offset for adaptation, which results in less offset misalignment and leads to a better generative model adaptation with directional loss.

## H    LIMITATION

Our proposed iterative refinement method seeks to improve the quality of zero-shot adaptation. However, it relies entirely on the pre-trained CLIP representation space, inheriting any biases and errors present in CLIP. Additionally, as noted by Guo et al. (2023), achieving adaptation across large domain gaps, such as "Human" to "Cat," is particularly challenging. Our approach necessitates that the trained generator closely approximate the target domain before initiating iterative refinement.

## I    SOCIAL IMPACT

The AIR methodology holds potential for enhancing artistic image synthesis in social media contexts and could serve as a beneficial data augmentation tool in other computer vision tasks such as recognition and detection. However, its capability to generate realistic images from real-world data raises ethical considerations. It is crucial to address these issues thoughtfully to prevent misuse and ensure responsible application of this technology.

## J    LICENSES

In Table 7, we specify the source and licenses of the models and datasets used in our work. Note that the FFHQ dataset consists of facial images collected from Flickr, which are under permissive licenses for non-commercial purposes.

