# OpenReview forum: "AIR: Zero-shot Generative Model Adaptation with Iterative Refinement"
_ICLR.cc/2025/Conference — ICLR 2025 Conference Withdrawn Submission_

### Official Review · Reviewer_pPcL · 2024-11-01

**Soundness:** 2
**Presentation:** 3
**Contribution:** 2
**Rating:** 5
**Confidence:** 3

**Summary:**

This paper presents a novel approach called AIR to address the zero-shot generative model adaptation (ZSGM) problem. First, it performs an empirical study to analyze the misalignment between text offsets and image offsets in CLIP. Second, it proposes AIR to address this issue.

**Strengths:**

1.The paper conducts an empirical study on a large public dataset to analyze offset misalignment in the CLIP embedding space, finding that misalignment increases as the concepts become more distant

2.Figures 2 and 3 vividly present the misalignment in the CLIP space and illustrate the impact of offset misalignment

**Weaknesses:**

1. There is a concern about the paper lacks theoretical proof or experimental evidence that the after limited iterations of the adaptation, the adapted generator is already closer to the target domain than the pre-trained generator

2. There is no sensitivity study conducted for the parameters t_int and t_thresh. Since these parameters play a critical role in introducing adaptive loss and updating the anchor points, their impact should be analyzed.

3. This paper lacks comparative experiments involving ITI-GEN, as the design of its learning prompt is based on ITI-GEN (Lines 353-357).

**Questions:**

Could you conduct an experiment or provide a theoretical proof for the statement: 'After a limited number of adaptation iterations using directional loss, the encoded concept in the adapted generator is already closer to the target domain than the encoded concept in the source generator'? This is an important assumption underlying your method.

---

### Official Review · Reviewer_eU1s · 2024-11-03

**Soundness:** 3
**Presentation:** 3
**Contribution:** 3
**Rating:** 8
**Confidence:** 3

**Summary:**

This paper explores zero-shot adaptation for generative models, particularly diffusion models. It empirically examines the offset misalignment issue in previous methods and the impact of this issue on generative model adaptation. The paper proposes an iterative refinement method to mitigate the effects of offset misalignment.

**Strengths:**

1. This paper is well-organized and clearly presented.
2. The study on offset misalignment is novel to me.
3. The iterative refinement solution for misalignment is interesting and reasonable.

**Weaknesses:**

I did not find any remarkable flaws in this paper. However, I have one question: in the study in Section 3.1, the concept distance is measured between different classes, whereas in the impact study in Section 3.2, the concept distance is constructed using different hand-crafted prompts. Why are these setups misaligned?

**Questions:**

Please refer to the Weakness section.

---

### Official Review · Reviewer_iT2P · 2024-11-04

**Soundness:** 3
**Presentation:** 3
**Contribution:** 3
**Rating:** 6
**Confidence:** 3

**Summary:**

In the ZSGM field, previous work has simply aligned the image offset with text offset using directional loss. However, experiments show that these offsets are not merely aligned but often misaligned. Based on this finding, the authors propose Adaptation with Iterative Refinement (AIR) to alleviate this issue by iteratively selecting anchor points closer to the target domain. The anchor points are selected during adaption, coupled with a new prompt learning approach.

**Strengths:**

1. This paper is well-written and clearly introduces its motivation and research methods. First, it highlights the limitation of previous methods, which simply aligned image offsets with text offsets, and verifies this limitation through experiments. Next, it conducts experiments to validate the hypothesis that addressing these misalignments can lead to improved performance. Finally, based on this analysis, the paper presents its proposed research method.

2. The paper conduct an analysis of the offset misalignment, and then the first to reveal the misalignment is larger for distance concepts and less for close concepts.

**Weaknesses:**

1. Table 1 and Table 2 present the results of the GAN model and the diffusion model, respectively. However, the evaluation metrics, comparison methods, and adaptations used for the two models are not consistent.

2. Most of the experiments conducted involve adaptation between two concepts with similar images.

3. Line 153 states, "Previous works assume that for two different concepts, α and β." However, Algorithm 1 and Algorithm 2 use two learning rates, also denoted as α and β. This could lead to confusion.

**Questions:**

1. Why are Table 1 and Table 2 not consistent? Please explain the differences in the evaluation metrics, comparison methods, and adaptations used for the GAN model and the diffusion model.

2. How does it perform on adaptation when there are significant differences between the source and target images? For example, in the NADA[1] experiment: Dog -> The Joker, Dog -> Nicolas Cage.

3. Note that Algorithm 1 and Algorithm 2 have two learning rates, α and β. Are these two learning rates consistent? How was the learning rate of 0.002 chosen in Supp. A.4?

[1] Rinon Gal, Or Patashnik, Haggai Maron, Amit H Bermano, Gal Chechik, and Daniel Cohen-Or. Stylegan-nada: Clip-guided domain adaptation of image generators. ACM Transactions on Graphics (TOG), 41(4):1–13, 2022.

**Details Of Ethics Concerns:**

The authors discuss the limitations and ethical issues that we are concerned about in Supp. I and J.

---

### Official Review · Reviewer_HwPH · 2024-11-04

**Soundness:** 2
**Presentation:** 2
**Contribution:** 2
**Rating:** 3
**Confidence:** 5

**Summary:**

This paper introduces an innovative Adaptation with Iterative Refinement (AIR) method for addressing offset misalignment in CLIP embedding space within Zero-Shot Generative Modeling (ZSGM). Through a detailed empirical study, the authors analyze the offset misalignment between image and text offsets in CLIP embedding space, demonstrating that this misalignment intensifies with greater concept distance, yet is less impactful between closer domains. To counter this, the AIR method iteratively samples anchor points during adaptation, utilizing a novel prompt learning strategy to describe these anchor points without predefined textual descriptions. The proposed approach effectively mitigates offset misalignment, resulting in good performance in ZSGM for diffusion models.

**Strengths:**

The paper introduces a novel approach to Zero-shot Generative Model Adaptation (ZSGM) by addressing the critical issue of offset misalignment between image and text representations in the CLIP embedding space, showcasing originality in its formulation and methodology.

**Weaknesses:**

1.	The references are somewhat disorganized and have formatting issues; for example, many citations should be formatted as (Smith et al., 2023) rather than Smith et al. (2023). Additionally, there is a lack of coherent context when citing references.
2.	The writing of this paper could benefit from some improvement, as it contains several spelling errors (e.g., "Adaptatoin" instead of "Adaptation") and some grammatical inconsistencies.
3.	In the Related Work section, this paper assumes that many methods default to the alignment of image and text offsets in CLIP space, which seems to warrant further consideration. For instance, some works, such as SVL, have already discussed T2I consistency. Furthermore, some studies have also addressed zero-shot content consistent T2I, such as Tewel, Yoad, et al. "Training-free consistent text-to-image generation."
4.  Lacking some more convincing qualitative and quantitative experiments (e.g., Figure 4, Figure 5), as well as a comparison of the diversity of entities.

**Questions:**

1.	In Figure 4 (left), there is no significant difference in qualitative results between AIR, NADA, and IPL. Could this indicate that, in single-descriptor, same-category adaptation, existing methods do not exhibit a significant T2I offset? Otherwise, please provide some more convincing qualitative experiments to support this.
2.	In Table 3, please explain why aligning offsets significantly improves generation diversity without causing the generated content from the model to become too similar, resulting in a decrease in diversity.
3.	Compared to animals and humans, the qualitative and quantitative experiments in the paper seem to lack content-consistent generation for some objects and scenes, as seen in other works. Please provide more diverse experimental examples and results to enhance the paper's persuasiveness.
4.	The paper seems to lack a qualitative ablation study. Please provide some specific experimental results to supplement it.
5.	If the text description in a sentence (rather than a specific prompt) is inherently ambiguous, is the method of aligning offsets presented in this paper still useful?
6.	Regarding the results in Figure 5, for example, the eyes in the photo → cartoon transformation seem to move, and a similar issue appears in the dog example. This problem also seems to be present in Figure 6 of the supplementary materials. Please provide more compelling results.

---

### Note · Authors · 2024-11-15

I have read and agree with the venue's withdrawal policy on behalf of myself and my co-authors.